# Design, Fabrication, and Application of Mini-Scaffolds for Cell Components in Tissue Engineering

**DOI:** 10.3390/polym14235068

**Published:** 2022-11-22

**Authors:** Vladimir A. Mironov, Fedor S. Senatov, Elizaveta V. Koudan, Frederico D. A. S. Pereira, Vladimir A. Kasyanov, Jose Mauro Granjeiro, Leandra Santos Baptista

**Affiliations:** 1Center for Biomedical Engineering, National University of Science and Technology “MISIS”, 119049 Moscow, Russia; 2Laboratory of Cell Technologies and Medical Genetics, National Medical Research Center for Traumatology and Orthopedics Named after N.N. Priorov, 127299 Moscow, Russia; 33D Bioprinting Solutions, 115409 Moscow, Russia; 4Joint Laboratory of Traumatology and Orthopaedics, Riga Stradins University, LV-1007 Riga, Latvia; 5Bioengineering Laboratory, National Institute of Metrology, Quality and Technology (INMETRO), Duque de Caxias 25.250-020, Brazil; 6Campus UFRJ Duque de Caxias Prof Geraldo Cidade, Universidade Federal do Rio de Janeiro, Duque de Caxias 25.240-005, Brazil

**Keywords:** mini-scaffolds, functionalization, tissue spheroids, synergistic approach

## Abstract

The concept of “lockyballs” or interlockable mini-scaffolds fabricated by two-photon polymerization from biodegradable polymers for the encagement of tissue spheroids and their delivery into the desired location in the human body has been recently introduced. In order to improve control of delivery, positioning, and assembly of mini-scaffolds with tissue spheroids inside, they must be functionalized. This review describes the design, fabrication, and functionalization of mini-scaffolds as well as perspectives on their application in tissue engineering for precisely controlled cell and mini-tissue delivery and patterning. The development of functionalized mini-scaffolds advances the original concept of “lockyballs” and opens exciting new prospectives for mini-scaffolds’ applications in tissue engineering and regenerative medicine and their eventual clinical translation.

## 1. Introduction

The conceptual basis of tissue engineering technology, since its inception, includes three pillars, or the so-called triad of tissue engineering: (i) cells, (ii) scaffolds, (iii) bioreactor-based chemical and/or physical signals [1]. The scaffold is a key element of this triad, and the scaffold could be defined as a temporal and removable (often synthetic or natural biodegradable polymer) support. The scaffold usually represents the so-called scaffold-based top-down approach in tissue engineering. Scaffolds, as temporal support, biodegrade before or after implantation and are replaced by a natural extracellular matrix synthesized by cells seeded on scaffolds [2]. Recently, the conventional scaffold-based approach in tissue engineering started to undergo several challenges or paradigm shifts. Firstly, the novel concept of scaffold-free tissue engineering has been introduced and has rapidly emerged as a potentially superior alternative to the classic scaffold-based approach. In the context of this paradigm shift, tissue spheroids or cell aggregates have been introduced as building blocks in tissue engineering and especially in bioprinting technology [3,4,5]. Secondly, the novel concept of self-assembled mini-scaffolds or “lockyballs”, based on the bottom-up approach, has been developed and successfully introduced in the tissue engineering field [6,7,8] (Figure 1). Lockable mini-scaffolds or “lockyballs” advance the original concept of scaffold-based tissue engineering or paradoxically open a potential combination of scaffold-based and scaffold-free approaches in tissue engineering with a potential synergetic approach [9,10]. The recent review explores the possible synergy of scaffold-based and novel scaffold-free approaches in tissue engineering [10].

This review aims to estimate the perspectives of applying a mini-scaffold-based approach in tissue engineering with special attention on their functionalization and use as cell delivery devices in regenerative medicine.

## 2. Third Strategy in Tissue Engineering

As we already indicated in the previous section of this review, there are two main strategies in tissue engineering: (i) scaffold-based and (ii) scaffold-free approaches. There are certain advantages and disadvantages to both approaches in tissue engineering [9,10]. The scaffold-based approach has a low cell density and obvious limits for self-assembly, whereas the scaffold-free approach has inferior initial material properties of tissue spheroids and limits for their functionalization. The combination of both approaches provides an obvious synergetic effect by the elimination of existing limitations of two different approaches in tissue engineering. Thus, the synergetic combination of two main different approaches represents a so-called third strategy in tissue engineering. We strongly believe that the third strategy in tissue engineering has a strong translational and commercialization potential, and it will eventually significantly advance the field of tissue engineering and regenerative medicine. One advantage of the combination of scaffold-free tissue spheroid-based approach and mini-scaffold-based approach is the advanced functionalization of tissue-engineered constructs. For example, magnetic functionalization will enable magnetic positioning and precision delivery of mini-scaffolds loaded with dense-packed cells. Specially designed hooks and loops in “lockyballs” enable interlocking and implementation of a bottom-up approach (Figure 1) [6,7,8]. Finally, a biodegradable mini-scaffold can also serve as a very cost-effective local drug delivery system for local delivery of growth and differentiation factors. Of course, design, fabrication, and especially functionalization of mini-scaffolds are not trivial tasks, and they will require expensive equipment such as two-photon polymerization devices and sophisticated photo-sensitive polymers. However, the potential clinical translation of the proposed synergetic approach or third strategy in tissue engineering is becoming increasingly obvious.

## 3. Creation and Characterization of Functionalized Mini-Scaffolds

### 3.1. Design of Mini-Scaffolds

One of the main challenges of the mini-scaffold-based approach is the design of mini-scaffolds. We will consider three cases. The design of so-called “lockyballs” with special hooks enabling interlockable self-assembly was not a trivial engineering task because both too many hooks and too few hooks prevent self-assembly. Optimal interlocking has been achieved only after using the so-called “elevated pentagon,” which serves as some sort of loop and enables effective interlocking of two closely placed lockyballs and, thus, rapid biofabrication of tissue-engineered constructs with a desirable patient-specific geometric shape (Figure 1) [7]. In this context, “lockyballs” serve as a variant of interlockable “Lego” blocks. It was very interesting that the replacement of “hooks” and “loops” employed originally in the design of classic biomimetic “Velcro” on simple arrows also enabled even more effective and more flexible interlocking (Figure 1). Arrows or arrow-headed spikes in this context serve both as classic hooks and loops in “Velcro”, simultaneously [8]. Moreover, the design of a spiral-like mini-scaffold with attached arrows will enable the rapid biofabrication of tubular tissue-engineered constructs of different diameters, wall thickness, and layered composition by combining a scaffold-free cell sheet technology with snake-like arrow-headed mini-scaffolds. For example, sequential attachment of several cell sheet monolayers with arrowed snake-like spiral mini-scaffold will enable rapid biofabrication of the tissue-engineered vascular graft with an endothelial cell-derived monolayer as an intima, smooth muscle cell-derived multilayer as a media, and fibroblast cell-derived multilayers as an adventitia. Finally, the combination of superior flexibility in the precision fabrication of the internal three-dimensional geometry of mini-scaffolds using two-photon polymerization and photo-sensitive polymers open unlimited possibilities for realizing the desirable advanced complex mini-scaffold design. Finally, the original unique design of a mini-scaffold for in situ robotic bioprinting of human hair has been recently developed (Figure 2).

### 3.2. Fabrication of Mini-Scaffolds

There are several different methods of fabrication of different mini-scaffolds. We will focus our attention on the description of the three most popular methods of mini-scaffold fabrication: (i) two-photon polymerization technology (Figure 1), (ii) sacrificial biomaterials, and (iii) melt electrospinning. Two-photon polymerization is a relatively expensive method for the fabrication of mini-scaffolds which, however, became popular due to its high resolution. The second limitation of this otherwise very promising method is the necessity to use specially designed and synthesized expensive photo-sensitive biodegradable and non-toxic polymers and hydrogels [11,12]. Additionally, two-photon polymerization does not allow simultaneous fabrication of mini-scaffolds encaging living cells; it is possible to do only after fabrication (Figure 3). Since its inception, the concept of using sacrificial materials such as sugar in tissue engineering for the fabrication of porous sponge-like mini-scaffolds is becoming increasingly popular. Biomaterials were selected for the fabrication of a mini-scaffold filled with densely packed biodegradable microspheres from sacrificial materials. The dissolving of densely packed and contacted sacrificial materials allows for the formation of a porous mini-scaffold. The porous structure of the resultant mini-scaffold enables sequential cell seeding (Figure 3). The different sacrificial materials, including crystals of salt and sugar, could be used for the fabrication of a mini-scaffold. However, the potential toxicity of sacrificial material is the most important limiting factor for this technology. Finally, rapidly evolving melt electrospinning is becoming another prospective method for precise high-resolution fabrication of mini-scaffolds for tissue engineering and regenerative medicine [13,14,15].

It is logical to assume that new emerging bioprinting methods, as well as other rapidly evolving technologies of biofabrication of mini-scaffolds loaded with tissue spheroids, will eventually be also successfully developed.

### 3.3. Cellularization of Mini-Scaffolds

The cellularization of mini-scaffolds is a critical, and probably the most important, step in the practical implementation of the so-called third strategy in tissue engineering. Usually, the cellularization of mini-scaffolds started after their fabrication using two-photon polymerization technology or other methods. The fabrication of mini-scaffolds and their simultaneous cellularization is a still unsolved technical issue due to the obvious toxic effect of recent methods of fabrication of mini-scaffolds. In this situation, it is logical to assume that the mini-scaffold with pores is suitable for seeding by living cells (Figure 3). However, the challenge is to perform cellularization with the maximal possible cell density, which will enable, for example, the biofabrication of tissue-engineered tissue spheroids encaged in a mini-scaffold and provide the necessary potential for tissue spheroid fusion. In this context, one of the most effective strategies for the cellularization of mini-scaffolds by living cells is using confined space provided by molded hydrogels (Figure 3). In this case, a mini-scaffold is placed in a confined place or recession fabricated from a molded hydrogel, and then this confined space is filled sequentially with a high-concentration suspension of living cells. The mathematical modeling and computer simulation using the open source software “Surface Evolver” [see http://facstaff.susqu.edu/brakke/evolver/evolver.html] demonstrated that during tissue spheroid formation, there is almost a 25% reduction in the initial diameter of the tissue spheroid due to so-called random dense cell packing, or simply tissue compaction (Figure 3) [7]. Thus, at least theoretically, it is possible to calculate optimal cell density and the desirable diameter of the tissue spheroid encaged, properly located, and positioned inside the mini-scaffold. The second strategy for the cellularization of mini-scaffolds is based on using tissue spheroid-based seeding. In this case, the mini-scaffold must have a port or entrance for inserting large diameter tissue spheroids capable of some sort of elastic squeezing during the insertion process with the sequential restoration of initial tissue spheroid shape and size but already inside mini-scaffolds.

### 3.4. Functionalization of Mini-Scaffolds

Functionalization of mini-scaffolds is the next logical step in the advancing of mini-scaffold technology in tissue engineering. We will consider here four main types of possible functionalization of mini-scaffolds: (i) enhancing mini-scaffold material properties; (ii) enhancing interlockability and self-assembly of the mini-scaffold; (3) magnetic functionalization for precision cell delivery and retention; (4) loading the mini-scaffold with drugs (growth and differentiation factors) and, thus, constructing a drug-eluting system or pharmacologically functionalized mini-scaffold. To enhance the material properties of mini-scaffold Russian dolls, “Matryoshka”-like designs have been employed. Basically, it consists of three interconnected balls inside one mini-scaffold. The resulting material properties of mini-scaffolds with the “Matryoshka”-like internal design (as it has been demonstrated by tensiometry) have been dramatically enhanced (Figure 1) [8]. The functionalization, by introducing the interlockability of mini-scaffolds, has been implemented either by using analogs of a “hooks” and “loops“ (elevated pentagon) system, which was similar to the classic “Velcro” design in so-called “lockyballs”, or by using arrowed interlockable spikes instead of hooks and loops (Figure 1). Finally, for magnetic functionalization, we initially planned to use two-photon polymerization of a photopolymer containing magnetic nanoparticles (superparamagnetic iron oxide nanoparticles or, shortly, SPION). However, at least theoretically, nanoparticles could interfere with two-photon polymerization. The second approach for effective functional magnetization of mini-scaffolds has been based on coating a fabricated mini-scaffold with magnetic materials. The advantage of magnetic functionalization was the absence of any possible interference with the two-photon polymerization process (Figure 4). Finally, the pharmacological functionalization of the mini-scaffold and fabrication of drug-eluting tissue-engineered scaffolds is also possible. In the case of mini-scaffold functionalization, the size of the scaffold does not matter too much.

### 3.5. Estimation of Material Properties of Mini-Scaffolds

The superior material properties of mini-scaffolds, as compared with the inferior material properties of tissue spheroids, have been long considered one of the main advantages of the scaffold-based approach in tissue engineering. Tissue spheroids can also significantly improve their biomechanical properties during cultivation in vitro, but this usually needs several weeks. We will consider here at least three special cases of estimation of material properties of fabricated mini-scaffolds: (i) scaffold tensiometry, (ii) spike flexibility, and (iii) scaffold retention in a skin test. Tensiometry is a classic method of biomechanical testing in which two parallel plates compress the object (in our case, mini-scaffold) with increasing levels of physical compressing force, and the resulting displacement is evaluated, so-called stress-strain curves are constructed, and based on these curves, modules of Jung are estimated (Figure 1j). For the effective performance of tensiometric material testing, the lockyballs must be fabricated without their hooks [8]. In the second case, the flexibility of a harpoon-like spike in a mini-scaffold for bioprinting of human hairs (so-called “capillinser” from Latin words: “capillo” = hair and “inserto” = to insert) has also been estimated using tensiometry (Figure 2). Finally, for estimating the retention in the skin of the inserted capillinser, special biomechanical testing must be performed. For such testing, the capillinser must be connected to a very sensitive mechanical sensor, and the force necessary and sufficient for the removal of the inserted capillinser from the human skin must be quantitatively estimated. The force which it is necessary to apply for the removal from the natural skin hair could serve as a control. The mathematical modeling and computer simulation technologies using finite element analysis (FEA) software such as “ANSYS” and its analogs will allow an estimate of material properties of the designed mini-scaffold in silico even before actual fabrication and mechanical testing [8] and, thus, the optimization of the design of the mini-scaffold will be enabled in silico.

## 4. Biomedical Applications of Mini-Scaffolds

### 4.1. Application of Magnetic Mini-Scaffolds for Precision Cell Delivery

One of the most exciting recent functionalizations of mini-scaffolds is their magnetization (Figure 4, Table 1). Magnetic mini-scaffolds have been used for the precision position in cell delivery, using a specially designed magnetic positioning system or simply a magnet [16,17,18,19,20]. The development of precision cell delivery using the magnetic functionalization of mini-scaffolds has been based on three important advances. Firstly, an effective method for magnetic functionalization, or simply magnetization, of fabricated mini-scaffolds has been developed. Secondly, magnets, or, in some cases, specially designed magnet-based positioning systems, have been developed. Finally, research groups developed several in vivo animal models, starting with a chick chorioallantoic system and zebrafish embryos and finishing with nude mice, have been employed [16,17,18,19,20]. The chick chorioallantoic membrane (CAM), as well as nude mice, do not have a functional immune system that allows use of a mini-scaffold loaded with human cells or the encasement of tissue spheroids from human cells. Taken together, these three important advances strongly indicate that magnetic functionalization of mini-scaffolds for cell delivery could be successfully commercialized and eventually clinically translated. The main advantage of magnetic mini-scaffolds compared to magnetic labeling of living cells and sequential biofabrication of magnetic tissue spheroids is that in the first case, there is an often unsolved issue of toxicity of nanoparticles, which impedes many attempts to translate nanotechnology-based technologies of cell delivery into clinical regenerative medicine. In the case of magnetically functionalized mini-scaffolds, the contact zone, or areas of direct contact of living cells with magnetic materials, is very limited, and magnetization of mini-scaffolds does not require their adhesion to the cell surface or even intracellular localization of magnetic nanoparticles. Moreover, magnetic-based cell and tissue delivery by employing magnetically functionalized mini-scaffolds also enables their potential long-term retention in the desired place in the human body.

### 4.2. Kensan Technology for Robotic Biofabrication of Tissue-Engineered Hair from Tissue Spheroids Inside Specially Designed Mini-Scaffold (“Capillinser”)

The original technology (so-called Kensan technology) based on using thin needles for patterning tissue spheroids and biofabrication of tissue-engineered constructs have been recently developed by Japanese bioengineers [29,30,31]. The developers claimed that it is a so-called scaffold-free technology. However, they are using metallic needles as temporal and removable (but not biodegradable) supports, which fits perfectly with the definition of the scaffold as a temporal and removable support. The mechanism of removal is not essential in the definition of the scaffold. The technology is very close to clinical translation, and it is already maximally automated. We proposed using the above mentioned Kensan technology for robotic seeding of duplets or pairs of tissue spheroids inside mini-scaffolds (“capillinsers”) for in situ bioprinting of tissue-engineered human hair [32]. As a first step, two types of tissue spheroids from epidermal cells (keratinocytes) and dermal cells (fibroblasts) are fabricated. In the second step, these spheroids are penetrated or punched (like shish kebab) by a thin metallic needle, at first spheroids from dermal fibroblasts and then spheroids from epidermal keratinocytes. A pair of closely placed tissue spheroids or duplets is critically important for the induction of differentiation processes leading to the morphogenesis of human hair. Finally, these punched pairs of tissue spheroids are robotically placed inside the mini-scaffold (“capillinser”). Due to the unique biomechanical properties of tissue spheroids (tissue spheroids, from the physical point of view, represent visco-elastic-plastic soft matter or some sort of complex fluid), they could be squeezed inside specially designed mini-scaffold “capillinsers” through the relatively narrow entrance. The final step is needle removal (Figure 5B). The whole procedure fabrication of spheroid duplets encaged in a mini-scaffold (“capillinser”) could be automated.

### 4.3. Fabrication of Tissue-Engineered Tubular Constructs Using Tissue Spheroids Seeded on Electrospun Matrices and Arrowed Helix-Like Mini-Scaffold (“Velix”)

Many human tissues and organs, from a geometrical point of view, are tubes. From an anatomical point of view, tubes are relatively simple in their construction. It is not surprising that many tissue engineers focus their attention on the biofabrication of tubular tissue-engineered constructs [33]. For example, there are many reviews on tissue-engineered blood vessels and vascular grafts [34,35,36]. However, existing methods of biofabrication of tissue-engineered constructs are usually time-consuming, and as a result, they are very expensive and commercially not feasible. Moreover, for the maturation of tubular tissue-engineered constructs, special perfusion bioreactors and expensive perfusion media are necessary. Thus, bioreactor-free technology for rapid and cost-effective biofabrication of tubular tissue-engineered constructs is highly desirable. We have already published a review about bioreactor-free technologies for tissue engineering of tubular tissue constructs. Electrospinning technology is one of the popular and relatively simple, and cost-effective methods of rapid fabrication of tubular scaffolds and tubular tissue-engineered constructs [37,38,39]. It has demonstrated that tissue spheroids could be placed and densely packed on electrospun matrices, and they can later attach and spread on electrospun matrices. A specially designed helix-like arrowed mini-scaffold or arrowed spiral, which we proposed to call “velix” (from a combination of two Greek words “velos” = arrow and “elix” = helix), could be considered as a type of long mini-scaffold (Figure 6). Fabrication of a layer of closely packed tissue spheroids on electrospun matrices will represent the first step in biofabrication, whereas rolling of the mini-scaffold “velix” with simultaneous penetration by arrows in this cellularized electrospun matrix will be the second and final step. Thus, the tubular tissue-engineered construct will be biofabricated.

### 4.4. Magnetic Nanoparticles as Mini-Scaffolds for Patterning of Tissue Spheroids

In conventional tissue engineering, cells are seeded on a scaffold. The rapid development of nanotechnology enables the biofabrication of mini-scaffold or temporal and removable supporting structures in tissue engineering using nanoparticles [40,41,42,43]. Nanoparticles could be used as temporal, removable, and often biodegradable supports, but placed inside cells. From the tissue engineering point of view, magnetic nanoparticles are the most interesting biomaterials which can serve as intracellular mini-scaffolds [44,45,46,47]. This very promising direction in tissue engineering has been called magnetic forces-driven tissue engineering [47,48,49,50]. Tissue spheroids could also be magnetized if they are fabricated from cells labeled with magnetic nanoparticles. Tissue spheroids fabricated from cells labeled with magnetic nanoparticles could be patterned and even bioassembled in assembloids using external magnets providing magnetic fields (Table 1). It has been demonstrated by several groups that tissue spheroids labeled with magnetic nanoparticles (such as SPION—superparamagnetic iron oxide nanoparticles) could be patterned in linear, circular, and branched patterns [21,22,23,40] (Figure 7). Magnetic nanoparticles are typical nanostructures, and from a formal point of view, they could not belong to a mini-scaffold. However, instead of nanoparticles, microparticles could be used. We recently reported successful patterning of tissue spheroids with magnetic microparticles with a size of several microns [24].

The main issue with using nano- and microparticles as mini-scaffolds is their potential cytotoxicity. From one point of view, they biodegrade to ferritin, which is a natural component of human blood, that can be removed with urea [51]. From another point of view, undesirable accumulation of magnetic nanoparticles in human organs, such as the spleen and liver, with the potential development of hemosiderosis has been reported. The concentration of magnetic nanoparticles and microparticles, which must be sufficient for their magnetic functionalization, is another important issue. Finally, technology must be cost-effective.

### 4.5. Bioprinting of Mini-Scaffolds for Tissue Spheroids

Bioprinting is a very promising technology for biofabrication of mini-scaffolds encaging tissue spheroids [52,53,54,55]. Bioprinting could be defined as a robotic additive biofabrication of functional tissue and organ constructs from living cells and biomaterials according to digital models [56]. There are three main methods for biofabrication of mini-scaffolds containing tissue spheroids: (i) bioprinting 3D tissue construction from hydrogel (or bioink) containing tissue spheroids; (ii) bioprinting of scaffolds from hydrogel and inserting tissue spheroids in the bioprinted scaffold holes; (iii) bioprinting of scaffolds from solid polymers and inserting tissue spheroids in bioprinted scaffold holes (Figure 8, Table 1). The first method is using analog or continuous dispensing or extrusion technology. The main challenge of the second and third methods is the development of a bioprinter capable of printing one spheroid at a time. There are already commercial bioprinters capable of printing one spheroid at a time.

A combination of biomaterials (hydrogels) and living cells is called a bioink. The development of bioinks is one of the most important trends in the development of bioprinting technology [57,58,59]. However, most existing bioink compositions are a combination of cell suspension with hydrogels. The maximum possible concentration of cells in a bioink is around 30%. Further increasing the cell concentration in a bioink will induce shear stress related to cell injuries and will produce an impediment to bioprinting. The thickness of bioprinted strands in standard fused deposition-based bioprinting technology is usually 400 μm. The optimal diameter of tissue spheroids is 250 μm. Thus, at least theoretically, it is possible to print bioinks containing tissue spheroids. Indeed, this has been achieved by De Moor et al. [25]. Three-dimensional bioprinting of mini-scaffolds from hydrogel containing tissue spheroids at first increases the viability of cells located in the center of tissue spheroids. Secondly, the embedding of tissue spheroids in hydrogel prevents the formation of a capsule-like barrier structure on the surface of tissue spheroids.

Another method of bioprinting of mini-scaffolds containing tissue spheroids is inserting tissue spheroids in the printed mini-scaffold from a hydrogel. It could be a porous mini-scaffold with pores suitable for the robotic insertion of tissue spheroids. Alternatively, tissue spheroids could be directly inserted into the continuous layer of a printed hydrogel.

In order to successfully implement this technology of bioprinting of mini-scaffolds, it is necessary to have at least several components: (i) printable hydrogel; (ii) tissue spheroids of standard size and shape; (iii) bioprinter capable of working in both analog and digital (one spheroid a time) modes of action; (iv) digital model.

One of the most important issues directly related to the bioprinting of tissue spheroids in hydrogel scaffold is enabling tissue spheroid fusion [60]. Ideally, the scaffold must permit tissue spheroid fusion both in vertical and horizontal directions. This could be accomplished by punching or robotic insertion of tissue spheroids in sequential printed layers of a hydrogel. Embedding tissue spheroids in the mini-scaffold hydrogel is permissive for tissue spheroid fusion in case of direct contact with an adjacent tissue spheroid embedded in a hydrogel. For successful implementation of vertical tissue spheroid fusion, the thickness of the printed layer of hydrogel must correspond to the diameter of embedded tissue spheroids. This will enable direct contact of the bioprinted tissue spheroid both in the horizontal and vertical direction and biofabrication after fusion of a 3D mini-tissue construct.

In the case of the second approach based on the robotic insertion of tissue spheroids into porous printed hydrogel mini-scaffold, there are certain challenges for tissue spheroid fusion in the horizontal direction. Maximal reduction of the thickness of printed strands that form a porous scaffold is one way to enable the horizontal tissue spheroid fusion process. Using a biodegradable hydrogel is another possible way to enable horizontal tissue spheroid fusion. Enhancing the retraction of mini-scaffold strands separating the inserted tissue spheroids also represents a possible way to enable horizontal tissue spheroid fusion. 

The third approach in the bioprinting of mini-scaffold containing tissue spheroids has been developed by Schon BS and Mekhileri NV et al. [26,27,61]. They used the classic 3D printing method—a fused deposition modeling (FDM) which allows the use of solid plastics for the fabrication of porous scaffolds. However, for inserting tissue spheroids into the porous scaffold, they use specially designed bioprinter model capable of inserting one tissue spheroid at a time. The main advantage of using a solid scaffold, which enables printing mini-scaffold with very thin strands, is that it does not interfere with horizontal tissue spheroid fusion. It has been demonstrated that chondrospheres fabricated from chondrocytes and robotically placed in a 3D mini-scaffold were able to fuse both in a horizontal and vertical direction even before the biodegradation of a printed mini-scaffold [27]. A robotic 3D bioprinter enables biofabrication of scaffolds seeded with tissue spheroids. However, technically it is also possible to insert tissue spheroids manually using so-called manual pipetting. For example, Laronda MM and co-authors, during biofabrication of a mouse ovary, used a 3D printer for the fabrication of a mini-scaffold from a collagen hydrogel and then used manual pipetting to insert tissue spheroids containing ovary tissue with ovarian follicles [28,62]. It is interesting that the functionality of biofabricated and genetically GFP-labeled ovarian tissue constructs have been confirmed by their transplantation into mice that underwent ovariectomy. Thus, a mini-scaffold containing tissue spheroids with viable and functional human ovarian follicles could be used for the treatment of naturally occurring and artificially induced infertility. Manual pipetting is indeed a very realistic and promising approach in tissue engineering of mini-scaffolds seeded with tissue spheroids, but it does not fit our definition of bioprinting as a truly robotic and not manual technology. Thus, 3D bioprinting is a rapidly evolving prospective approach for the biofabrication of mini-scaffolds containing tissue spheroids. The development of 3D bioprinters capable of printing tissue spheroids (one tissue spheroid at a time) is probably the single most important advance in this emerging field. It is safe to predict that bioprinting will be a popular method for the biofabrication of tissue spheroids embedded in mini-scaffolds.

### 4.6. Biofabrication Using Mini-Scaffolds and Tissue Spheroids as a “Bottom-Up” Approach in Tissue Engineering

Conventional tissue engineering is based on the so-called “top-down” approach when large organ-size scaffolds are seeded with cells [63,64]. The obvious limitations of such an approach are several problems with cell seeding on large scaffolds: (i) after initial cell seeding of the surface area of a large size scaffold, the next cell seeding becomes limited; (ii) after seeding of large size scaffolds, the central part of tissue engineered constructs are subject of hypoxia and associated cells death; (iii) vascularization of large size tissue engineered constructs based on using a large size scaffold is still an unsolved problem and a long-standing challenge in tissue engineering; finally, (iv) positioning and placing of different cell types inside a 3D scaffold according to the histo- and organo-typical pattern is a really challenging and not trivial task.

Biofabrication of tissue-engineered constructs using tissue spheroids encaged into mini-scaffolds represents an alternative approach. It is based on the so-called “bottom-up” approach. The possibility of in situ biofabrication of patient-specific shaped tissue-engineered constructs is one of the main advantages of this approach. Rapid in situ assembly of tissue-engineered constructs from tissue spheroids encaged in mini-scaffolds also enables layered positioning of different cell types. Thus, it constitutes the second advantage of the bottom-up approach. Finally, the bottom-up approach using pre-vascularized tissue spheroids [65,66] allows the resolution of the problem of vascularization and the viability of tissue-engineered constructs.

Therefore, the application of mini-scaffolds encaging tissue spheroids, or the bottom-up approach, allows for solving some long-standing unsolved problems of a conventional top-down approach in tissue engineering and provides certain advantages and much more flexibility in the biofabrication of histotypical tissue-engineered constructs [67,68,69].

## 5. Challenges of Clinical Translations of Mini-Scaffold Technology

The application of mini-scaffolds for cell delivery and rapid biofabrication in tissue engineering and regenerative medicine faces several challenges [16,17,18,19,20] (Figure 4B and Figure 5A). First, ideal biomaterials for the fabrication of mini-scaffolds must be processable, biocompatible, and biodegradable. As we mentioned already, the best method for the fabrication of mini-scaffolds needs both specially designed, and FDA- (or by some other national regulatory agency) approved biodegradable and non-toxic, suitable for sterilization, photo-sensitive polymers or biomaterials, as well as an expensive two-photon polymerization device also officially certified for clinical used. Second, the optimal cell source is still an unsolved issue in tissue engineering in general. Ideally, we must have an unlimited number of autologous cells which will not be able to induce an undesirable immune response in the form of immune rejection and foreign body reaction. They also must have strong regenerative potential and an intrinsic capacity to proliferate, differentiate and produce an authentic extracellular matrix. Third, aside from special devices for the fabrication of mini-scaffolds per se, we must also have certified for clinical use devices for rapid biofabrication, such as 3D bioprinters, biofabricators, and bioassemblers, as well as special mini-scaffold precision delivery and positioning devices, such as, for example, magnetic positioners in the case of using magnetically functionalized mini-scaffolds. Finally, there is an obvious need for adequate pre-clinical in vivo animal models for predictable testing of the effectiveness of cell delivery using mini-scaffolds, their biocompatibility and biodegradability, and clinical suitability and efficacy as a novel therapeutic modality. There is a growing consensus that regulatory approval for using mini-scaffolds in clinical practice by correspondent regulatory agencies is probably the single most challenging issue and even a potential impediment to clinical translation and sequential commercialization of promising mini-scaffold technology in tissue engineering and regenerative medicine.

## 6. Conclusions

Mini-scaffolds are small scaffolds of different shapes with sizes ranging from one hundred micrometers to several millimeters. The application of mini-scaffolds represents a rapidly evolving and promising prospective approach for cell delivery and rapid biofabrication of 3D tissue-engineered constructs in tissue engineering and regenerative medicine. The synergetic combination of the scaffold-free approach with the mini-scaffold-based approach opens an exciting prospective of a potentially more superior so-called third strategy in tissue engineering, which combines the advantages of both scaffold-free and scaffold-based approaches. Further progress in the design, fabrication, and especially in the functionalization of mini-scaffolds will advance both cell delivery and rapid biofabrication in clinical tissue engineering and regenerative medicine.

## Figures and Tables

**Figure 1 polymers-14-05068-f001:**
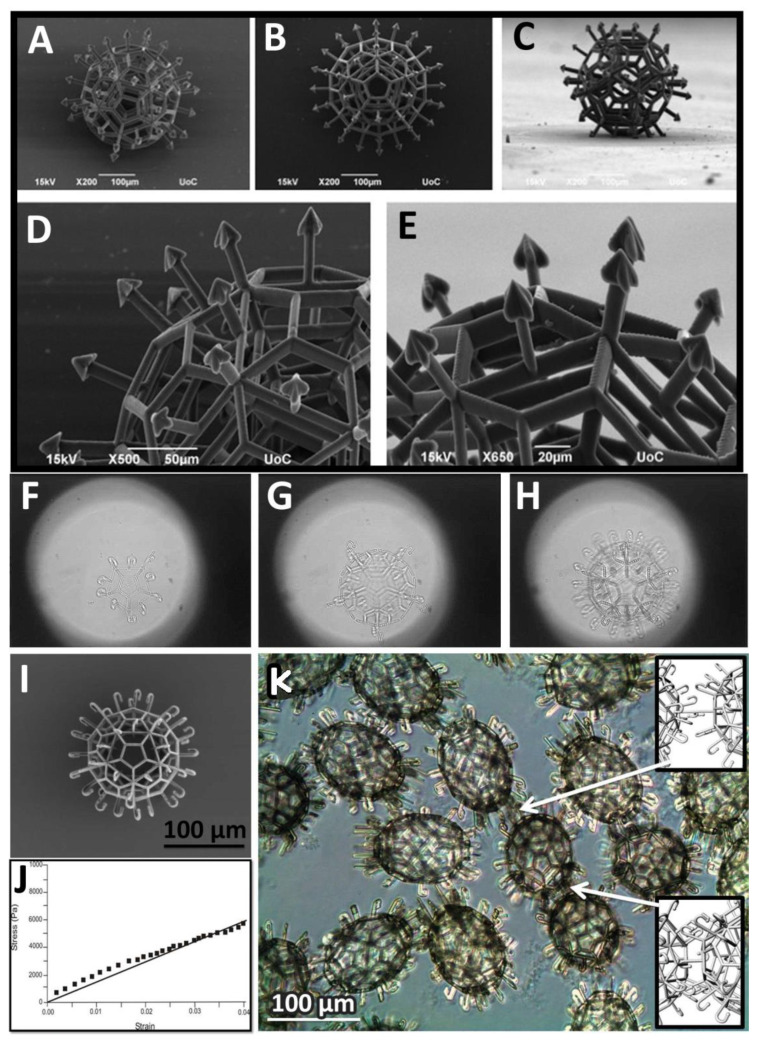
Design, fabrication by two-photon polymerization, sequential material characterization, and estimation of interlockability of mini-scaffolds (“lockyballs”). (**A**–**E**) SEM of arrow-headed “lockyballs” (Reprinted with permission from [8]. Copyright [2015], American Vacuum Society); (**F**–**H**) sequential steps of layer-by-layer fabrication of “lockyballs”; (**I**) SEM of “lockyball”; (**J**) material properties of “lockyballs”—stress-strain curve; (**K**) interlocking mechanism of “lockyballs” (**F**–**K** reprinted from [7]).

**Figure 2 polymers-14-05068-f002:**
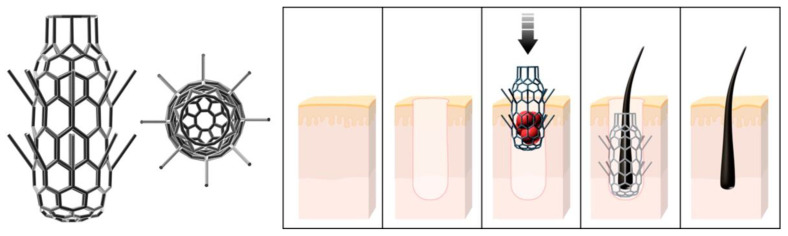
Design of mini-scaffold (“capillinser”) and scheme of sequential steps of robotic in situ bioprinting of tissue-engineered human hairs in the human skin using a specially designed mini-scaffold (“capillinser’).

**Figure 3 polymers-14-05068-f003:**
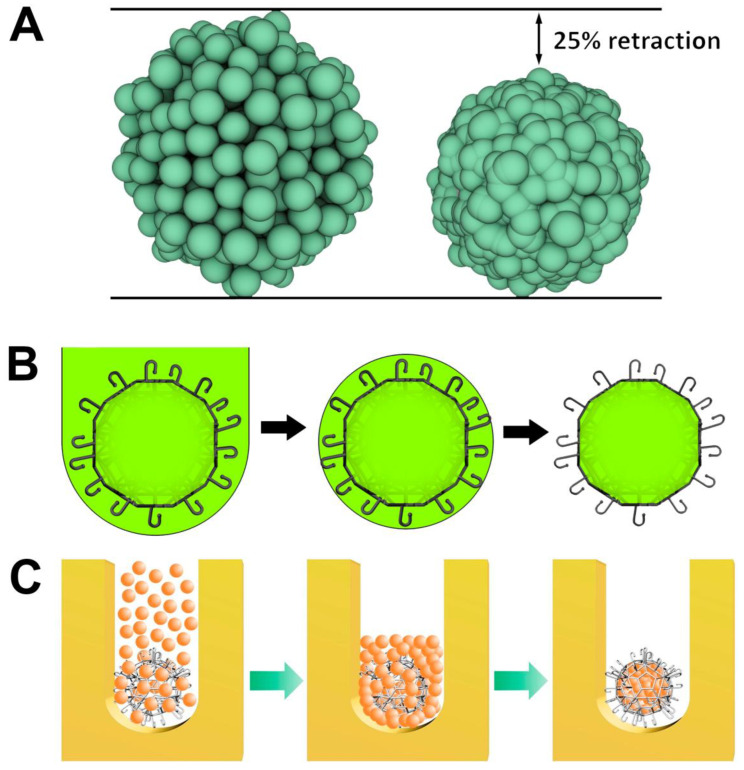
Mechanism of cell seeding and sequential tissue compaction in mini-scaffolds. (**A**) Mathematical modeling and computer simulation of tissue spheroid compaction using the open source software “Surface Evolver”. (**B**) Estimation of potential redistribution of cells around the mini-scaffold during tissue spheroid compaction. (**C**) Three sequential steps of cell seeding and tissue spheroid compaction around the mini-scaffold (Reprinted from [7]).

**Figure 4 polymers-14-05068-f004:**
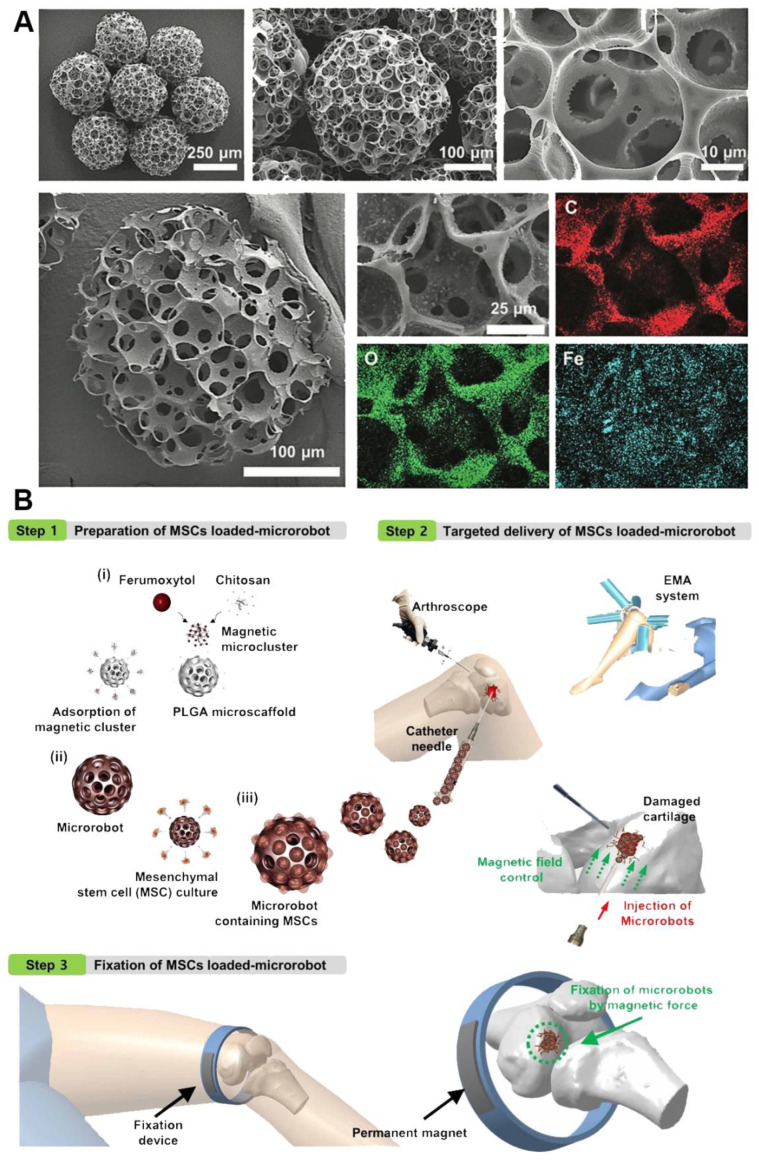
(**A**) Magnetic mini-scaffold for cell delivery fabricated using sacrificial materials. (**B**) Concept overview of knee cartilage regeneration procedures using a magnetic microrobot-mediated mesenchymal stem cell delivery system (From [16]. Reprinted with permission from AAAS).

**Figure 5 polymers-14-05068-f005:**
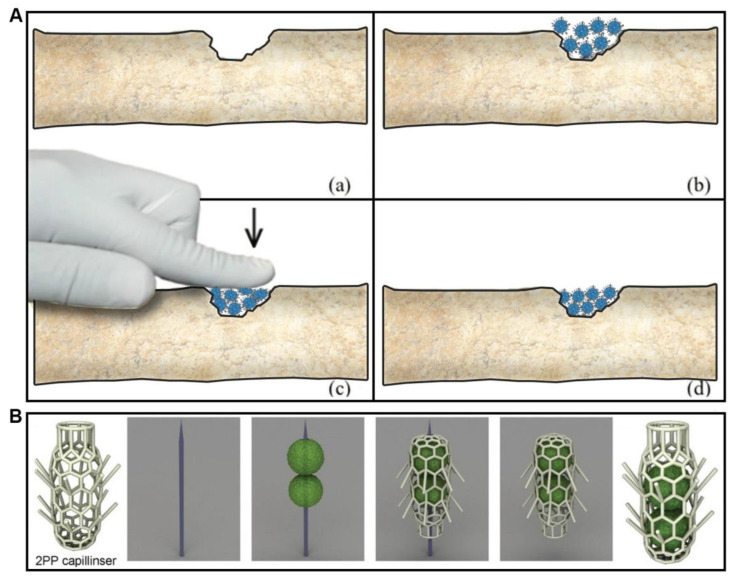
(**A**) Schematic of in situ rapid biofabrication of an osteogenic tissue-engineered construct using tissue spheroids encaged into interlockable concentric lockyballs (Reprinted with permission from [8]. Copyright [2015], American Vacuum Society); (**B**) Insertion of pairs of tissue spheroids into the mini-scaffold “capillinser” (Reprinted from [32]).

**Figure 6 polymers-14-05068-f006:**
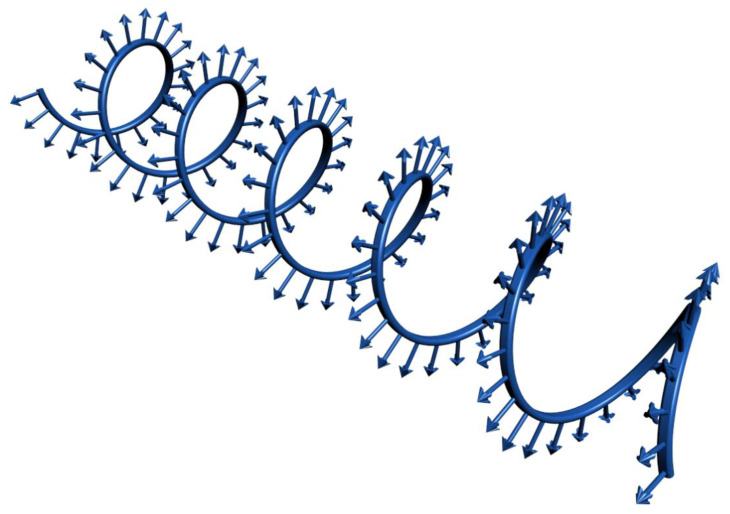
“Velix”—a spiral shape arrowed mini-scaffold for rapid biofabrication of tubular tissue-engineered constructs.

**Figure 7 polymers-14-05068-f007:**
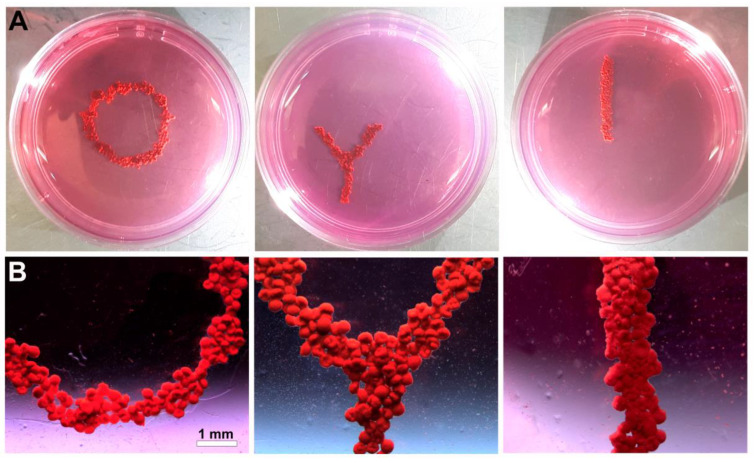
Forming circular, branched, and linear constructs by magnetic patterning of tissue spheroids labeled with magnetic nanoparticles. (**A**) Photographs and (**B**) stereo images of tissue constructs.

**Figure 8 polymers-14-05068-f008:**
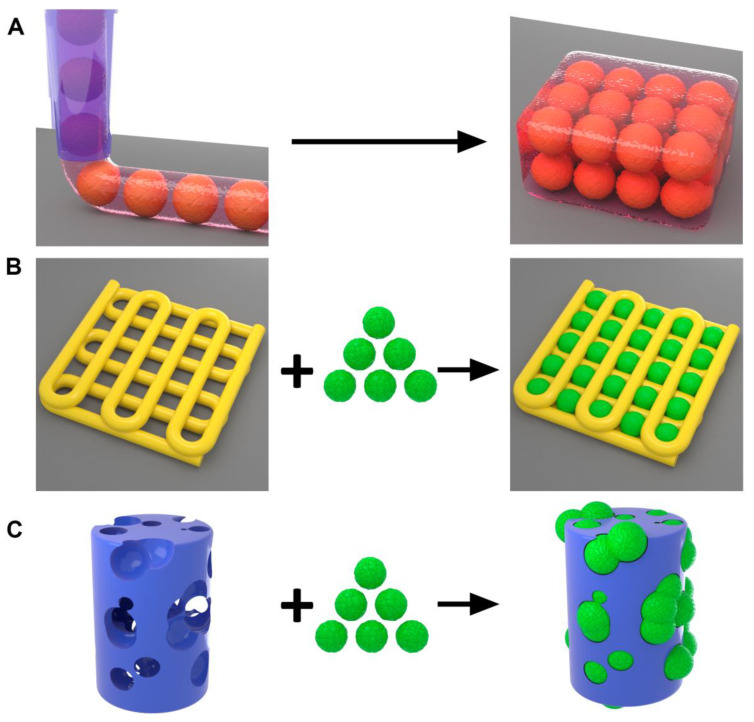
Three methods for biofabrication of mini-scaffolds containing tissue spheroids. (**A**) Bioprinting 3D tissue construction from hydrogel (or bioink) containing tissue spheroids. (**B**) Bioprinting of scaffolds from hydrogel and inserting tissue spheroids in the bioprinted scaffold holes. (**C**) Bioprinting of scaffolds from solid polymers and inserting tissue spheroids in bioprinted scaffold holes.

**Table 1 polymers-14-05068-t001:** Biomedical applications of mini-scaffolds.

Application	Type of Mini-Scaffold	In Vivo Model or Type of Tissue Construct	Reference
Precision cell delivery	SPIONs (superparamagnetic iron oxide nanoparticles)	Athymic BALB-C nude mice, brain tissue	[16]
Precision cell delivery	Magnetic microbot	Nude mice, liver tissue	[17]
Precision cell delivery	Magnetic microbot	Balb/C nude mice, zebrafish embryos	[19]
Precision cell delivery	Magnetic microbot	Rabbit, knee cartilage	[20]
Magnetic patterning of tissue spheroids	Magnetic monosized polymer microspheres Dynabeads M-450 functionalized with carboxyl groups (Invitrogen)	Small ring, large ring, hexagon, triangle, and arrays from HepG2 spheroids	[21]
Magnetic patterning of tissue spheroids	Streptavidin MagneSpheres paramagnetic particles (Promega)	Linear construct from HeLa spheroids	[22]
Magnetic patterning of tissue spheroids	SPIONs	Branched construct from endothelial cell spheroids	[23]
Magnetic patterning of tissue spheroids	Magnetic microcapsules	Linear, circular, and branched constructs from NIH3T3 spheroids	[24]
Bioprinting of tissue-engineered constructs	Hydrogel with tissue spheroids	Cartilage microtissue from photocrosslinked gelatin methacryloyl (GelMA) hydrogel with chondrogenically induced human mesenchymal stem cell spheroids	[25]
Bioprinting of tissue-engineered constructs	Solid polymers	Cartilage construct based on scaffold from bio-degradable poly(ethylene glycol)-terephthalate—poly (butylene terephthalate) block co-polymers and spheroids from human chondrocytes	[26,27]
Bioprinting of tissue-engineered constructs	Hydrogel	Ovarian construct based on scaffold from gelatin hydrogel and ovarian follicles	[28]

## Data Availability

Not applicable.

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
