# Peer review of "Design, Fabrication, and Application of Mini-Scaffolds for Cell Components in Tissue Engineering"

_polymers, 2022, doi:10.3390/polym14235068_

Round 1

Reviewer 1 Report

Very interesting paper. Concerns mini-scaffolds for tissue engineering and regenerative medicine. Introduction well analyzed on the basis of articles. The conducted research and conclusions prove the very good knowledge of the subject by the authors. 

Author Response

We are very thankful to the reviewers for careful reading of the manuscript!

Reviewer 2 Report

In the review paper, the authors summarized the design, fabrication and application of 3D hollow microstructures for regenerative medicine. Generally, the flow of this story is fine, however, there are some major issues needs to be clarified:

1, the definition: if the locky ball structure could be considered as 3D microscaffolds, how about the liposomes (GUVs), giant polymersomes? Therefore, the concept of mini-scaffolds should be well proposed, with comparison from other papers.

2, The structure of this paper should be well arranged, rather than from section 1 to 14. Should be outlined like, introduction, fabrication techniques, applications, and outlook.

3, Figure 1-3 should be combined, since they are used to describe one story. In the meanwhile, the scale bar in Figure 1-2 are unclear or unknown, which should be revised.

4, Figure 6-7 should be combined into one image; also applied to figure 8 and 9; And the image quality should be enhanced as well.

5, The current section 11- 13 should be provided with some more images, to supporting the discussion.

6, there are too few discussion with the application in regenerative medicine, as mentioned in the title, which should be improved.

7, The language is really poor, with some many grammar mistakes, thus needing well polishing by some language agency.

Author Response

We are very thankful to the reviewer for careful reading of the manuscript!

1, the definition: if the locky ball structure could be considered as 3D microscaffolds, how about the liposomes (GUVs), giant polymersomes? Therefore, the concept of mini-scaffolds should be well proposed, with comparison from other papers.

Author’s reply:

We thank the reviewer for the consideration of our manuscript. The idea of locky balls as 3D microscaffolds was in formation of tailored-architectured structure. In this regard, these structures were identified as a separate group, rather than such carriers as liposomes and giant polymersomes.

2, The structure of this paper should be well arranged, rather than from section 1 to 14. Should be outlined like, introduction, fabrication techniques, applications, and outlook.

Author’s reply:

We have rearranged the manuscript structure.

3, Figure 1-3 should be combined, since they are used to describe one story. In the meanwhile, the scale bar in Figure 1-2 are unclear or unknown, which should be revised.

Author’s reply:

We have combined figures 1 and 3 (Figure 1, page 2) and added the scale bar. Figure 2 has been excluded.

4, Figure 6-7 should be combined into one image; also applied to figure 8 and 9; And the image quality should be enhanced as well.

Author’s reply:

We have combined figures 6 and 7 (Figure 4, page 7) and figures 8 and 9 (Figure 5, page 9).

5, The current section 11-13 should be provided with some more images, to supporting the discussion.

Author’s reply:

We have added two additional images (Figure 7, page 11 and Figure 8, page 12) to this section.

6, there are too few discussion with the application in regenerative medicine, as mentioned in the title, which should be improved.

Author’s reply:

The title has been changed to “Design, fabrication, and application of mini-scaffolds for tissue engineering”.

7, The language is really poor, with some many grammar mistakes, thus needing well polishing by some language agency.

Author’s reply:

The whole manuscript has been thoroughly checked by a native speaker.

Round 2

Reviewer 2 Report

The authors addressed most of the questions, and the quality of this paper has been improved obviously. But two more questions are still unclear:

1, although the authors emphasized that mini scaffolds are different from GUVs or polymersomes, due to scaffolds are tailored; while, GUVs are also tailored with different building blocks, different structures, different loading capacities, and same functions as scaffolds, etc. So, in this sense, the reviewer can not fully agree with the authors. Please provide more explanation or adjust the title with more specific content.

2, One more suggestion for the improvement : please add few tables to provide a clear summary for the materials for mini-scaffolds, the size, the fabrication technique, the representative applications, etc. 

The language still needs improvement.

Author Response

1, although the authors emphasized that mini scaffolds are different from GUVs or polymersomes, due to scaffolds are tailored; while, GUVs are also tailored with different building blocks, different structures, different loading capacities, and same functions as scaffolds, etc. So, in this sense, the reviewer can not fully agree with the authors. Please provide more explanation or adjust the title with more specific content.

We thank reviewer for careful reading of the revised manuscript! 
We have adjusted the title as: "Design, fabrication, and application of mini-scaffolds for cell component in tissue engineering"

2, One more suggestion for the improvement : please add few tables to provide a clear summary for the materials for mini-scaffolds, the size, the fabrication technique, the representative applications, etc. 

According to the reviewer's suggestion we have added the table about biomedical applications of mini-scaffolds, which includes information about applications, types of mini-scaffolds and in vivo model or type of tissue 
construct.

The language still needs improvement.

The manuscript was checked by native speaker